# Neural Networks Applied for Predictive Parameters Analysis of the Refill Friction Stir Spot Welding Process of 6061-T6 Aluminum Alloy Plates

**DOI:** 10.3390/ma16134519

**Published:** 2023-06-21

**Authors:** Dan Cătălin Bîrsan, Viorel Păunoiu, Virgil Gabriel Teodor

**Affiliations:** Faculty of Engineering, Department of Manufacturing Engineering, “Dunărea de Jos” University of Galati, Domnească Street, 47, RO-800008 Galati, Romania; dan.birsan@ugal.ro (D.C.B.); virgil.teodor@ugal.ro (V.G.T.)

**Keywords:** stir spot welding, numerical simulation, aluminum alloy

## Abstract

Refill friction stir spot welding (RFSSW) technology is a solid-state joint that can replace conventional welding or riveting processes in aerospace applications. The quality of the new welding process is directly influenced by the welding parameters selected. A finite element analysis was performed to understand the complexity of the thermomechanical phenomena during this welding process, validated by controlled experiments. An optimization model using neural networks was developed based on 98 parameter sets resulting from changing 3 welding parameters, namely pin penetration depth, pin rotation speed, and retention time. Ten parameter sets were used to verify the learning results of the optimization model. The 10 results were drawn to correspond to a uniform distribution over the training domain, with the aim of avoiding areas that might have contained distortions. The maximum temperature and normal stress reached at the end of the welding process were considered output data.

## 1. Introduction

In 2002, GKSS-GmbH Germany [1] developed the refill friction stir spot welding process (RFSSW), which is a variant of the spot friction welding process (FSSW), the difference being that in the case of RFSSW, following the joint, there is no hole remaining in the sheet. 

The rotary tool consists of an external stationary clamping ring, a sleeve, and a rotating pin. The clamping ring has the role of fixing the two metal sheets. The sleeve can move vertically independently of the pin, which can rotate as well as translate vertically. The process consists of four stages. Stage 1 is the touchdown and preheating; the welding head clamps the two metal sheets using the clamping ring, sleeve, and rotating pin, thus, creating localized heat. Stage 2 is refilling; the sleeve advances downwards in sync with the pin retracting, resulting in material being redistributed. Stage 3 is retreating; the rotating pin moves in sync forward as the sleeve translates in the opposite direction. Stage 4 is the welding assembly retreating from the joint materials [2]. The stages are shown in Figure 1.

This welding process is suitable to be used successfully in the aeronautical, automotive, or naval industry, with the resulting welded joints being more resistant compared to those obtained by classical methods. One of the significant advantages of RFSSW technology is its ability to join dissimilar metals that cannot be joined by traditional welding methods. This means that manufacturers can use a variety of metals with different properties and characteristics, such as lightweight aluminum alloys, high-strength steels, and even copper alloys, to create complex structures and components that meet specific design requirements. This also leads to a reduction in weight, increased fuel efficiency, and improved performance in various industries, such as aerospace, automotive, and marine.

RFSSW technology also produces high-quality, defect-free welds with low distortion and minimal residual stress. This results in improved structural integrity and durability of the welded components, leading to increased product reliability and longevity. Additionally, RFSSW technology could be automated, which means that it can be easily integrated into existing manufacturing processes, resulting in higher production rates, reduced labor costs, and improved productivity.

Another significant advantage of RFSSW technology is its environmental sustainability. The process does not require any consumable materials, such as welding wire or filler material, which reduces material waste and eliminates the need for toxic coatings and adhesives used in other joining processes. RFSSW also produces no fumes, radiation, or other harmful byproducts, making it a safer and more environmentally friendly process.

In summary, the importance of RFSSW technology lies in its ability to join dissimilar metals, produce high-quality, defect-free welds, reduce distortion and residual stress, increase structural integrity and durability, improve product reliability and longevity, integrate with existing manufacturing processes, and promote environmental sustainability. These advantages make RFSSW technology an attractive option for a wide range of industries looking to improve their manufacturing processes and product performance.

A large number of studies were dedicated both to experimental work and to analyze the process using finite element modeling of thermo-physical phenomena that occur during the welding and experimental work.

The application of the finite element method (FEM) in simulation studies of RFSSW technology can be categorized into two distinct areas. The first category involves modeling the manufacturing process of a welded joint, encompassing technological analysis [3]. These simulations aim to understand various aspects of the welding process, such as friction phenomena, heat generation, material mixing, changes in material structure and physical state, and the distribution of residual stresses and deformations in the joint region [3]. The second category focuses on modeling welded joints as structural elements in loaded structures. These simulations primarily examine the behavior of a point joint under static or dynamic loads, [3].

Andres et al. [4] presents results of investigations on the RFSSW lap joints welded using rotations in the range from 1500 to 2000 rpm and tool sleeve plunge depths from 1.6 to 1.8 mm. The proposed experiment was to study the effects of selected parameters of the RFSSW process and surface preparation of joined elements on the microstructure and mechanical properties of lap joints.

Kubit et al. studied the effect of welding parameters on the load capacity and defects of welded single-lap joints of AA7075-T6 material [5]. The test results indicate that incorrect selection of welding process parameters results in defects, such as voids, hooks, onion rings, and bonding ligament errors.

The study of Shen, Z., et al. [6] focused on examining material flow, intermixing, and liquation cracking in dissimilar Al alloys during the refill friction stir spot welding (refill FSSW) process, employing a grooved tool. The findings unveiled that the welding procedure involved a combination of back extrusion and forging, and the vertical material flow contributed to the formation of a robust mechanical interlocking joint mechanism.

Zou, Y [7] conducted a study to examine the impact of rotation speed and plunge depth on the macro/microstructures and mechanical properties of dissimilar 2195/2219 aluminum alloy joints produced through refill friction stir spot welding (RFSSW). They observed that the change in rotation speed does not noticeably affect the occurrence of hook defects, whereas an increase in plunge depth leads to a significant increase in the height of these defects. Furthermore, the authors discovered a correlation between higher rotation speeds and increased tensile/shear strength. However, as the plunge depth increases, the tensile/shear strength initially rises and then decreases.

D’Urso et al. [8] performed experimental study of the FSSW process for the lap-joining of thin AA6061-T6 sheets. The FSSW process was applied on pairs of overlapped sheets by varying the tool rotational speed, and by keeping the other process parameters fixed, such as axial feed rate, indentation depth, and dwell time. The numerical results were validated by experiments. 

The objective of Andrzej, K. et al.’s [9] investigation was to assess how different parameters of refill friction stir spot welding (RFSSW) affect the fracture load and failure mechanisms of the resulting joints. The authors conducted RFSSW on 7075-T6 Alclad aluminum alloy sheets using various welding parameters and determined the load capacity of the joints under tensile/shear loading. To simulate the joint-loading process, they performed numerical simulations based on finite element analysis, considering the variation in elasto-plastic properties of the weld material across the joint cross-section.

Muci-Küchler, K.H, et al. [10] proposed a fully coupled thermomechanical finite element model of the refill friction stir spot welding (RFSSW) process of 6061-T6 aluminum alloy to predict the material flow, deformation, stress and strain distributions, and temperature distribution in the weld region over all phases of the process. The validation of the finite element analysis of the process was carried out by comparing the results obtained with those obtained from experimental investigations.

The material flow during the FSSW process of sheets of AA6061 was analyzed by Cao et al. [11], who used the Abaqus program to perform this simulation. The scope of this research was to study the formation of the “hook’ defect and onion structure near the periphery of the sleeve by modifying various parameters, such as temperature, stress, deformation, and deformation rate. 

Kluz, R et al. [12] presents an optimization method for the RFSSW process parameters to ensure maximum load capacity of the RFSSW joint. The study shows that tool rotational speed has the greatest impact on the load capacity of RFSSW joints, while high rotational speed adversely affects the weld structure. Increasing the duration of welding time results in significantly better weld properties with a smaller number of defects. However, the increases in the load capacity of the weld obtained by extending the duration of welding time are limited, leading to unfavorable phenomena in the HAZ manifested by a sharp reduction in the load capacity of joints. The study emphasizes the need for compromise solution in the selection of tool rotational speed and welding duration to ensure adequate plasticization of the base material, required load capacity of the joint, and appropriate joint microstructure. This study concludes that RFSSW is a successful method for joining aluminum alloys in the aircraft industry, reducing labor consumption in the process of the assembly of welded structures, as in Figure 2.

Janga, V.S.R et al. [13] presents a 3D numerical model for the refill FSSW process, employing an updated Lagrangian formulation through DEFORM-3D. The model accurately deals with severe element distortion through extensive remeshing. The numerically obtained temperature distribution is in good agreement with the experimental results, showing symmetric distribution about the weld center with the maximum temperature slightly above the reported solidus temperature. The distribution of plastic strain is also symmetric, where the maximum plastic strains occur around the shoulder region. The material flow analysis shows significant movement of plasticized material in the SZ, with intermixing of the material between the top and bottom sheets around the shoulder periphery. The maximum strain rates are observed around the shoulder, and the simulation results allow for distinguishing the SZ, TMAZ, and HAZ. The hardness profile correlates to the temperatures obtained from the simulation, with a hardness reduction in the zones. 

Effertz PS. et al. [14] aimed to optimize the refill friction stir spot welding process parameters using a multivariate polynomial regression (MPR) machine learning algorithm. The study found that the model exhibited significant dependence on the quadratic features of the studied parameters, with welding time and rotational speed having a detrimental effect on the ultimate lap shear force (ULSF). An optimum parameter combination was obtained within the investigated parameter window. Although the predicted optimum point did not correspond to the maximum obtainable ULSF, the microstructure exhibited clearly distinct zones found in stir based-processes besides a defect-free joint, highlighting the reliability of the parameters employed. While the results are promising, further research is needed to determine the adequacy of the model beyond the parameter window investigated and to compare other ML algorithms to the present solution.

Enkhsaikhan Boldsaikhan et al. [15] explores the use of neural networks (NN) for identifying weld quality in real-time and investigates the relationship between feedback force oscillations and material flow in friction stir welding (FSW). The study resulted in the development of an optimized NN with 60 input units, 9 hidden units, and 1 output unit, which correctly classified 95% of the testing set. A validation experiment was conducted to demonstrate the generality of the NN for classifying weld quality. The study suggests that the algorithm has potential for devising an intelligent feedback control system for friction stir welding (FSW), but further investigations are recommended to examine the impact of other factors on feedback forces, such as FSW machine variation, weld tool geometry, and the gap between workpieces.

Kubit, A et al. [16] present a fully coupled thermomechanical numerical model of the refill friction stir spot welding (RFSSW) process in Alclad 7075T6 aluminum alloy sheets. The model is based on finite element analysis and is validated by experimental results. The study shows that the material flow in the FE-based model is consistent with the experimentally fabricated joint. The temperature in the weld zone increases rapidly during the plunging stage and reaches its maximum at the point where the pin front meets the inner edge of the sleeve. The study also shows that the clad acts as a thermal insulator and inhibits homogeneous heat transfer in the weld, making the process difficult to control.

An axi-symmetric 2D model of the rotary friction stir spot welding (RFFSW) process, which successfully captures its physics, was developed by Berger, E. et al. [17,18]. The model accurately predicts heat generation and temperature distribution during welding, material flow, and defect formation in the weld material. However, it needs improvement to accurately predict the forces experienced by the shoulder. The study concludes that with more accurate flow stress data, the model could be a useful tool for optimizing the RFFSW process by estimating the optimal welding duration while achieving appropriate temperature evolution for good bonding.

In order to optimize the process parameters for refill friction stir spot welding of AA6082-T6 aluminum alloy, Silva, B.H. et al. [19] conducted a study to assess the individual influence of rotational speed, feeding rate, and plunge depth on weld strength. Statistical analysis was employed to maximize the peel and shear strength of the welds, revealing a notable correlation between plunge depth and weld resistance. The investigation highlighted the significant impact of plunge depth on the formation and shape of the hook defect, with the fracture behavior of the welds being highly dependent on the configuration of this defect.

Adamus, J. et al. [20] provide a comprehensive examination and categorization of defects found in aluminum joints created through refill friction stir spot welding (RFSSW) technology. The study presents the different failure modes observed, including fracture around the weld in both the top and bottom sheets, as well as rupture between the sheets. The progression of defects at the sleeve–plunger interface under deteriorating process conditions is also depicted. Finally, a comparison is made between RFSSW joints and the riveted joints currently employed in airplane structures. 

Takeoka, N. et al. [21] developed a new approach called scrubbing refill FSSW (Sc-RFSSW) which combines tool contact with the lower plate and material refilling. In Sc-RFSSW, the plunging depth of the shoulder is increased compared to conventional RFSSW. Their study focused on establishing the relationship between process parameters and the maximum joint load in an Sc-RFSSW joint consisting of an aluminum alloy and non-coated mild steel. Design of experiments (DoE) was employed to analyze the results. The DoE findings revealed that reducing the tool plunging depth into the lower plate, allowing it to only scrub the interface, led to an improvement in the maximum joint load. The optimized Sc-RFSSW joining condition consistently achieved a stable nugget pull-out in both tensile/shear and cross-tension tests.

Donga, Z. et al. [22] investigated the influence of joining time on the microstructure and mechanical properties of the (RFSSW) welded joints. The findings revealed that the Al/Mg alloys were effectively joined through the formation of a liquid eutectic phase induced by the stirring action of the tool. Following welding, an evident layer of intermetallic compounds (IMCs) developed at the center of the joint. At the pin-affected zone, an IMC layer with a thickness of approximately 1100 μm was observed, while the sleeve-affected zone exhibited a diffusion layer of only 10 μm. Interestingly, at a welding time of 1 s, minimal liquid eutectic phase was formed, resulting in a thinner 10 μm IMC layer at the pin-affected zone. However, as the welding time increased, thicker IMC layers were formed. The findings provide valuable insights into optimizing the RFSSW process for Al/Mg alloy joints.

Liu, X. et al. [23] developed a new approach called hybrid refill friction stir spot welding (RFSSW) combined with ultrasonic oscillation, which was introduced for joining 5A06 aluminum alloy. The study compared the metallographic structure and mechanical properties of RFSSW joints formed without ultrasonic assistance, with lateral ultrasonic assistance, and with longitudinal ultrasonic assistance. Furthermore, the process parameters for ultrasonic-assisted RFSSW were optimized.

Schmal et al. [24] provide an overview of refill friction stir spot welding (RFSSW) of high strength aluminum alloys, focusing on the boundary conditions and quality criteria of the welded joints. The influence of process parameters on joint formation and load-bearing capacities is discussed, considering the potential application of RFSSW in automotive and airplane production.

Zhao, Y. et al. [25] investigated the temperature and material flow behavior during friction stir welding of Alclad 7B04-T74 aluminum alloy these were studied by both numerical simulation and welding experiment. 

Song, M. et al. [26] presented a three-dimensional heat transfer model for friction stir welding (FSW). A moving coordinate was introduced to reduce the difficulty of modeling the moving tool and heat input from the tool shoulder and the tool pin were considered in the model.

Artificial neural network (ANN) analysis is used in refill friction stir welding (RFSSW) for several reasons. Some of the main reasons why ANN analysis is employed in RFSSW are as follows:

Process optimization: ANN analysis helps optimize the RFSSW process by modeling the complex relationships between process parameters and the resulting weld quality. By training an ANN with historical process data, it can learn to predict optimal parameter settings that lead to desirable weld characteristics, such as strength, hardness, and defect minimization.

Prediction of weld quality: ANNs are utilized to predict the quality of weld joints in RFSSW. By analyzing various input parameters, such as rotational speed, traverse speed, tool geometry, material properties, and environmental conditions, the trained ANN can estimate the resulting weld quality attributes. This enables operators to make informed decisions and take corrective actions to improve weld quality.

Real-time monitoring and control: ANNs can be integrated into RFSSW systems to provide real-time monitoring and control. By continuously analyzing sensor data during the welding process, the ANN can detect anomalies, identify potential defects, and provide feedback to adjust process parameters or alert operators if any deviations occur. This helps ensure consistent weld quality and reduces the likelihood of producing defective welds.

Parameter selection: ANNs assist in selecting appropriate process parameters for specific welding tasks. By training the network on a dataset that includes various combinations of process parameters and their corresponding outcomes, the ANN can learn to recommend optimal parameter sets for achieving desired weld characteristics. This helps save time and resources by guiding operators in selecting suitable parameters without the need for extensive trial-and-error experimentation.

Process monitoring and diagnostics: ANNs can be employed for process monitoring and diagnostics in RFSSW. By analyzing sensor data and historical information, the network can detect patterns and trends that indicate the health of the welding process. They can identify potential issues, such as tool wear, material inconsistencies, or improper process conditions, allowing for timely intervention and maintenance to prevent quality defects.

Overall, the use of ANN analysis in refill friction stir welding enhances process efficiency, improves weld quality, and enables better control and monitoring of the welding process. It leverages the power of machine learning to extract valuable insights from data, enabling optimization and automation in RFSSW operations.

Based on the above analysis, the paper proposes to develop a predictive model, based on neural networks, which can accurately predict the welding quality based on input parameters, such as the material properties, tool geometry, and process conditions. This can help identify the optimal process parameters for achieving the desired welding quality, while minimizing defects and process time.

## 2. Materials and Methods

The experiments for studying the effects of welding process parameters on the thermal distribution in the welding region on the mechanical properties of RFSSW lap joints were performed on 6061-T6 aluminum alloy sheets with a thickness of 1.5 mm.

The sheets of 80 × 20 × 1.5 mm (L × W × T) and 80 × 20 × 1.5 mm were overlapped and welded on a machine tool using a cylindrical tool. The welding tool used consists of three independent elements (Figure 1): a 22 mm diameter blank-holder, a 9.2 mm diameter sleeve, and a 5.8 mm diameter pin. The outer diameters of the pin and sleeve are 5.8 and 9.2 mm, respectively.

### 2.1. Methodology of the Welding Process

The finite element analysis was validated by analyzing temperature distribution and material flow behavior during experiments.

The RFSSW process has four main stages: touchdown and preheating (Figure 1a), when the tool starts to rotate to preheat the material of the upper sheet after the touchdown stage, plunging (Figure 1b), when the rotating sleeve plunges into the metal, while the pin moves upwards, refilling (Figure 1c), when the sleeve and pin reverse their direction and return back to their original position, and the retreating phase (Figure 1d).

Figure 3a illustrates the dimensions of the parts to be welded, while Figure 3b depicts the shape and dimensions of the tool used in the welding process.

The following welding parameters have been used in experiments: Clamping force of 17 kN (measured with a force transducer type HBM-1-C2/50KN);Tool rotational speed of 2100 rpm;Tool plunge depth of 1.45 mm.

The welding cycle consists of plunge time (1.2 s) and tool retracts time (1.2 s).

The metallographic analysis was performed after the welding process, with the piece being polished and analyzed microscopically to highlight the size and shape of the material grains. 

The chemical composition of aluminum alloy 6061-T6 is shown in Table 1.

### 2.2. Temperature Measurement

The temperature was measured during the welding process using two type K thermocouples connected to the HBM data acquisition device, type QuantumX-MX440B, Germany.

The thermocouples were mounted on the upper valve of the top plate, outside the sleeve radius, at a distance of 4.8 and 6 mm from the center of the pin, as shown in the schematic in Figure 4.

The recordings were made at 200 ms intervals. In order to obtain the most accurate results, 10 specimens were tested, and the average values of the measurements were used to validate the finite element analysis.

### 2.3. Finite Element Modeling

A finite element model has been developed to better understand the material behavior at the separation interface plane between the two welded materials, the flow of material, the temperature distribution, the stress and strains developed during the welding process, and the welding defects. An axisymmetric 2D analysis was performed due to the symmetrical nature of the process and to substantially reduce the analysis time. 

Thermomechanical coupled analysis was performed in the dedicated Simufact software (Simufact Engineering Gmbh, Hamburg, Germany). 

Depending on the real conditions, a number of simplifications of the numerical model were adopted without altering the obtained results. The sleeve, pin, and snap ring were all considered as rigid bodies. The rotational speed of the tool and its displacements were considered in agreement with the experimental conditions (see Figure 5).

The plates were modeled with 2D elements, intended for the analysis of 2D axiometric problems, called Quad (10) in Simufact Forming terminology, and the rigid tools were modeled with quadric elements called Quad (40) [27]. Using the advanced front Quad mesher, the initial mesh was generated (Figure 6) which, due to its exaggerated distortion during the welding process, leads to a divergence problem. To avoid this, an automatic rescaling was used, which automatically regenerates the meshes, and the simulation is continued using the new mesh. The upper and lower plates were modeled with 2300 elements together, and all rigid bodies (pin, sleeve, table, and hollow support) were modeled with 4079 elements.

The material properties (Table 2) were used from the material database included in the Simufact Forming software [27], and represent a temperature- and strain rate-dependent material model according to the MatiLDa database, which is a registered trademark of Gesellschaft für metallurgische Technologie-und Softwareentwicklung mbH (Berlin, Germany). 

The mathematical model of the material used is given by the following equation:(1)$$\sigma_{F}=C_{1}\cdot e^{\left(C_{2}\cdot T\right)}\cdot \varepsilon_{p}^{\left(n_{1}\cdot T+n_{2}\right)}\cdot e^{\left(\frac{l_{1}\cdot T+l_{2}}{\varphi }\right)}\cdot {\dot{\varepsilon }}_{p}^{\left(m_{1}\cdot T+m_{2}\right)}$$
where T is temperature, ε_p_ is plastic strain, ${\dot{\varepsilon }}_{p}$ is the plastic strain rate, and C_1_, C_2_, n_1_, n_2_, l_1_, l_2_, m_1_, and m_2_ are parameters which are determined based on the experimental data fitted by the plasticity model in Equation (1). 

The values of the parameters in the plasticity model are listed in Table 3. The Simufact Forming software does not provide a procedure for the determination of these parameters.

The equations used to analyze the friction stir welding process are as follows.

-The equation for the analysis of heat transfer during the process is as follows:

(2)$$\frac{\partial T}{\partial t}=\frac{\partial }{\partial x}\left(k_{x}\frac{\partial T}{\partial x}\right)+\frac{\partial }{\partial y}\left(k_{y}\frac{\partial T}{\partial y}\right)+\frac{\partial }{\partial z}\left(k_{z}\frac{\partial T}{\partial z}\right)+{\dot{q}}_{p}$$
where T is temperature, $\dot{q}$_p_ is the generation of heat from the dissipation of plastic energy, and x, y, and z are the spatial coordinates.

-For determining the rate of heat generation due to plastic energy dissipation, the following equation is used:

(3)$${\dot{q}}_{p}=\tau \cdot \eta \cdot {\dot{\varepsilon }}_{p}$$
where τ is the shear stress and η is the factor of conversion of mechanical to thermal energy.

-To determine the heat generated by friction between the tool surfaces and the workpiece, the following equation is used:

(4)$${\dot{q}}_{f}=\mu \cdot p\cdot \omega$$
where $\dot{q}$_f_ is the heat generated by friction, μ is the coefficient of friction, p is the contact pressure, and 𝜔 is the rotational speed.

-The following equation is used for determining the convective heat loss:

(5)$$q_{c}=h_{f}\left(T_{s}-T_{\infty }\right)$$
where h_f_ is the convection coefficient (h_f_ = 50 W/m^2^ °C), T_s_ is the temperature at the plate surfaces, and T_∞_ is the absolute temperature of the surroundings.

-For heat loss by radiation, the following equation is used:

(6)$$q_{r}=k\cdot \varepsilon_{r}\left(T_{r}^{4}-T_{\infty }^{4}\right)$$
where κ is the Stefan–Boltzmann constant (κ = 5.67·10–8 W/m^2^ °C), T_r_ is the absolute temperature of the radiating surface, and ε_r_ is the emissivity of the radiating surface.

The minimum moment required to rotate the circular tool relative to the workpiece surface in the plunge phase is as follows: (7)$$M=\int_{0}^{R}2\cdot \pi \cdot \mu \cdot F\cdot r^{2}dr$$
where R is the radius of the contact surface and F is the force.

Variations in physical–mechanical characteristics of the material, Young’s modulus, Poisson’s ratio, coefficient of thermal expansion, conductivity coefficients, and heat capacity under the influence of temperature were considered in the material database of the Simufact Forming software [27]. 

The value of the friction factor was modified according to the temperature (Figure 7), in accordance with the study of Zhao et al. [25] and Song et al. [26].

## 3. Results and Discussion

### 3.1. Thermomechanical Results of the RFSSW Process 

In the cross-section, Figure 8, the four zones, characterized by different microstructures, can be seen: the mixing zone (SZ), the thermomechanical affected zone (TMAZ), the heat-affected zone (HAZ), and the base material (BM).

#### 3.1.1. Temperature Distribution

The heat generated during the welding process is due to friction and plastic deformation in the welded area. Figure 9 shows a comparison between the temperature obtained from the finite element analysis and that recorded by the temperature sensors mounted on the top plate.

The initial temperature for the numerical model was 27 °C, the same as that recorded in the hall during the welding process.

In the first stage, which lasts 1.2 s, when the sleeve goes down and the pin goes up, the temperature rises very quickly in the welding zone. The maximum temperature recorded by sensor_1 was 172 °C at the time point of 2.1 s from the beginning of the welding process, while for sensor_2, the maximum temperature was 146 °C at 2.2 s (see Figure 9 and Figure 10).

The temperature rises very quickly in the welding zone to 310 °C/s; the maximum temperature value obtained during the welding process was 494 °C at a time of 1.8 s and, as can be seen in Figure 11, this occurs at the point where the pine front encounters the inner edge of the sleeve (Figure 12). 

The points selected to illustrate the temperature variation over time is presented in Figure 13. The graphs in Figure 14 and Figure 15 represent the temperature variation over time at the points located at the interfaces between the pin and the top plate and between the sleeve and the top plate.

#### 3.1.2. Mechanical Behavior of the RFSSW Process

Figure 16 and Figure 17 show the effective stresses and plastic deformations for both the plunge stage at 1.2 s from the start of the welding and the refill stage. Figure 17 presents the effective stresses and plastic deformations at the end of the refilling stage.

The maximum value of plastic deformation is 21 and is recorded at the end of the welding process.

The maximum effective stress has the value of 306 MPa, and this occurs at the interface between pin and sleeve, while the adjacent area of the upper plate material is further characterized by a fine-grained structure. 

The vectors in Figure 16c show us how the flow of material occurs in the joint for the two phases of the welding process.

In the plunge phase (Figure 16), one can see a deformation of the aluminum plates in the pin area, with the material tending to reflow in that direction, and a compression of the material in the sleeve area.

The typical defects of the RFSSW joint are incomplete filling (Figure 9) and structural notches in the base material of the upper sheet.

The reason for an incomplete refill could be that the material, following plastic deformation, reflows in the horizontal plane, as in this direction it has no dimensional constraints. Another factor could be the difference in the plasticity of the material in this area compared to the material in the area underneath the sleeve, due to the different friction between the tool and the workpiece in these areas.

The size of the structural notch depends to a large extent on the immersion depth of the tool, and this size has a considerable influence on the load capacity of the joint. In the proposed finite element model, but also in the welded part, for the parameters used, this type of defect was not recorded, as can be seen in Figure 9.

### 3.2. Neural Networks Applied for Predictive Parameters Analysis

In order to determine the influence of the parameters of the welding regime on the temperatures and stresses developed in the joint, an ANOVA type analysis was carried out using neural networks.

As is known, neural networks are based on the mathematical simulation of the neurobiological processes involved in the learning activity [28]. They are known for their ability to learn based on a multitude of presented situations and to synthesize the mathematical model of the proposed problem to be solved.

This ability of the neural network to learn by example gives them great applicability potential, especially in the area of solving complex problems [28,29].

It is enough to provide the network with a consistent set of examples simultaneously with a rule to modify the interneuron weights, and the network will compare the calculated output value with the output value given by the example. At the same time, the modification of the weights of the input data is also determined, in accordance with a previously specified strategy [30].

These aspects determine that the use of neural networks in ANOVA-type analyzes is very appropriate [31].

In order to carry out the mentioned analysis, RFSSW-type welding processes were simulated under the conditions of changing the following parameters of the welding regime:-The penetration depth of the pin. It was modified according to the values of 1.4, 1.45, and 1.5 mm.-The rotation speed of the pin. It was varied in the range of 1800 ÷ 2300 rpm, in steps of 100 rpm, obtaining six distinct situations.-Retention time. It was modified in steps of 0.2 s, in the range of 2 ÷ 3 s, obtaining six situations.

By changing the 3 parameters, a number of 6 × 3 × 6 = 108 examples were obtained.

Of these, 98 examples were used to train the neural network and 10 examples were kept for verifying the learning results.

The 10 results were drawn to correspond to a uniform distribution over the training domain, with the aim of avoiding areas that might have contained distortions.

The maximum temperatures, the temperature reached at the end of the welding process, the maximum normal stress, and the normal stress at the end of the process were considered as output data.

The JustNN program, version 4a, made by Neural Planner Software Ltd., was used to build and train the neural network.

The neural network training algorithm used by the JustNN program involves importing the data and changing its value within the existing limits. The output values are then predicted by the neural network and a new network is created using the new input data set and the predicted output data set. In the next step, the two networks are compared, and the input set is adjusted. The process continues until the new network provides an output data set that does not differ from the original set by more than the imposed error.

The program used (JustNN) does not allow modification of the default network training algorithm.

With the data obtained through simulation, a neural network was built with three hidden layers. The number of hidden layers was selected according to the recommendations of the specialized literature, which states that this number should be calculated with the formula $n=\sqrt{i\cdot o}$, with i being the number of input variables and o that of the output variables. In our case, the number of input variables is 3 (pin penetration depth s; pin speed n, and the holding time t) and the number of output variables is 4 (the temperature at the end of the welding process, T_end_; the maximum temperature reached during the welding process, T_max_; the stress reached at the end of the process, σ_end_; the maximum stress reached during the process, σ_max_). With these values, we obtained the value of 3.46, which we rounded to three hidden layers.

Figure 18 shows the diagram of the constructed network.

The values of the control elements for training the network were left at the default values of the JusNN program. These are the learning rate of 0.6, the momentum of 0.8, the first validation after 100 cycles, and the next validations after 100 cycles each. The condition for the learning stop was imposed so that all errors were less than 0.001, that is, each of the differences, in absolute value, between the initial value of the input parameter and the network-predicted value of the same parameter, was below the imposed limit.

After training the network, the importance of each of the input variables on the output variables was determined. In descending order of importance, they were pin penetration depth (s), having an importance of 28.743; pin speed (n), having an importance of 27.626; the holding time (t), having an importance of 17.871.

The input importance view is shown in Figure 19 and presents the importance and the relative importance of each input column. The importance is the sum of the absolute weights of the connections from the input node to all the nodes in the first hidden layer. The inputs are shown in the descending order of importance from the most important input.

It can be noted that the importance of the penetration depth and the rotation speed of the pin are almost equal, while the importance of the holding time is significantly less.

Figure 20 shows the graph obtained at the end of neural network training. One can observe the learning curves corresponding to the maximum, average, and minimum errors, respectively, the maximum allowable value of the error of which was set at 0.001. This value was reached after about 2420 learning cycles.

To verify and validate the obtained results, 10 network queries were made, based on the data kept for validation. These data were not used during the network training stage.

The values obtained were compared with the values obtained by simulation, and the results are presented in Table 4.

The meanings of the parameters in Table 4 are as follows: T_end_—the temperature at the end of the welding process; T_max_—the maximum temperature reached during the welding process; σ_end_—the stress reached at the end of the process; σ_max_—the maximum stress reached during the process; T_end_ calc—temperatures at the end of the welding process, predicted by querying the neural network; T_max_ calc—the maximum temperature reached during the welding process predicted by querying the neural network; σ_end_ calc—the stress reached at the end of the process obtained by interrogating the neural network; σ_max_ calculates the maximum stress reached during the process, obtained by querying the neural network.

The relative error was calculated for each of the output parameters with the following formula: $∆V=\frac{V_{s}-V_{c}}{V_{s}}\cdot 100$ [%], with vs being the value of each output parameter (T_end_, T_max_, σ_end_, σ_max_) obtained by modeling and V_C_ being the value of the same parameter obtained by querying the network.

It can be noted that the values obtained by querying the network are very close to the values obtained by simulation.

This allows us to state that the trained network can be used to make a prediction on the values of temperatures and stresses obtained under given conditions of pin penetration, rotation speed, and respective holding time.

The advantage over obtaining these values by simulation is that the response time of the neural network is much shorter and, also, the computational resources required are much lower compared to those needed for simulation.

A shortcoming of this predictive analysis is that the values obtained by querying the network are only realistic if the input values are in the range of values within which the training was carried out (but not necessarily equal to the training values).

## 4. Conclusions

-The temperature in the welding zone increases by 310 °C/s during the plunge phase; then, in the refill phase, there is a decrease in the first phase followed by an increase of about 700 °C/s for 0.3 s. In these conditions, the maximum temperature reaches 497 °C.-The temperature evolution in time, recorded both from thermocouples and from finite element analysis, is very close in shape and recorded values, with the correlation rate of the data exceeding 90%.-The deformations of the pieces subjected to the welding process and the flow of material in the joining area are very similar.-The importance of welding parameters, in decreasing order, is as follows: pin penetration depth (s), pin speed (n), and the holding time (t).-A trained neural network can be used to make a prediction on the values of temperatures and stresses obtained under given conditions of pin penetration, rotation speed, and respective holding time.

As for future research, we intend to focus on the following directions: -Research on grain size evolution in the context of refill friction stir spot welding (RFSSW).-Finite element modeling of the RFSSW process for dissimilar material joining.

## Figures and Tables

**Figure 1 materials-16-04519-f001:**
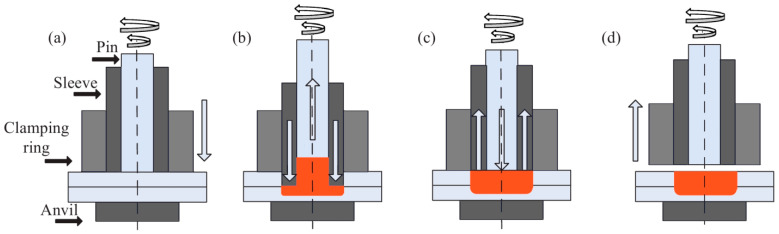
Stages of the RFSSW process: touchdown and preheating (**a**), plunging (**b**), refilling (**c**), and retreating (**d**).

**Figure 2 materials-16-04519-f002:**
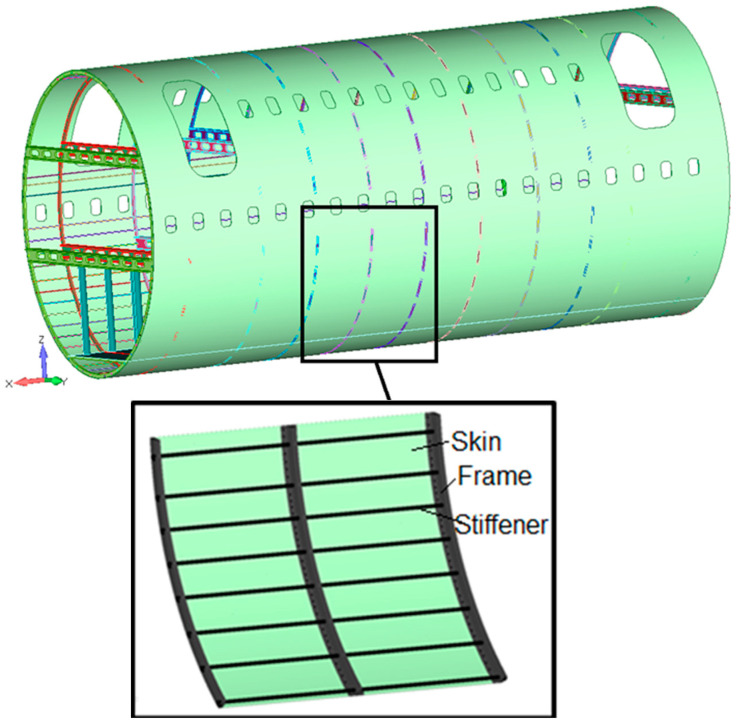
The structure of the fuselage.

**Figure 3 materials-16-04519-f003:**
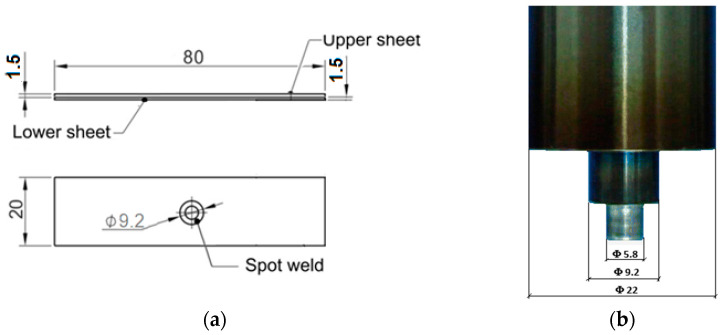
Shape and dimensions (in mm) of the welding plates (**a**) and tool (**b**).

**Figure 4 materials-16-04519-f004:**
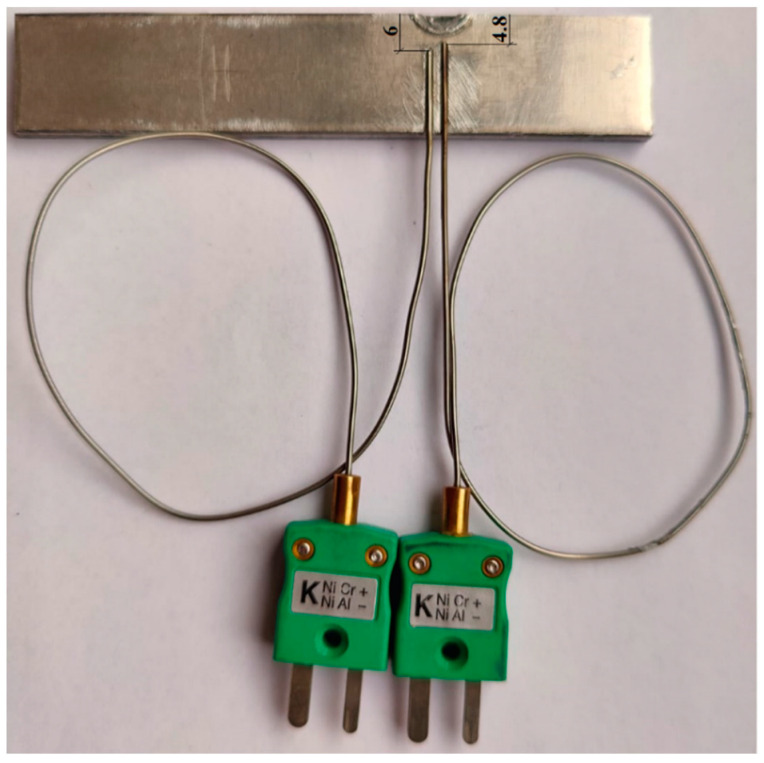
The wiring diagram for the thermocouples.

**Figure 5 materials-16-04519-f005:**
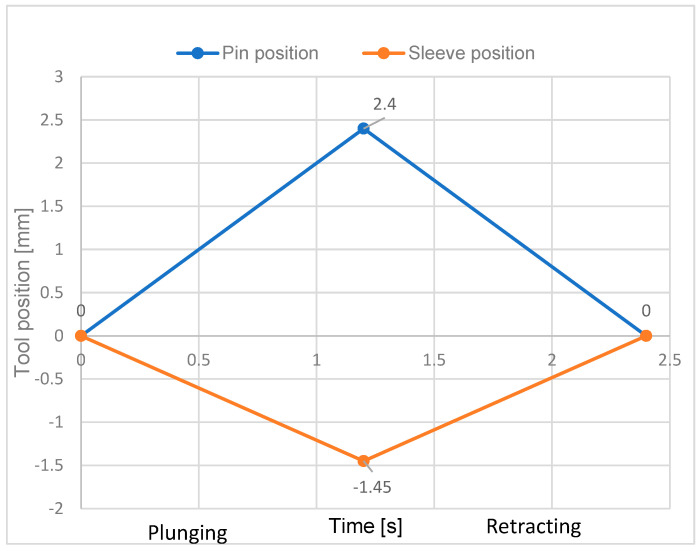
Stages of the RFSSW process considered in the numerical model.

**Figure 6 materials-16-04519-f006:**
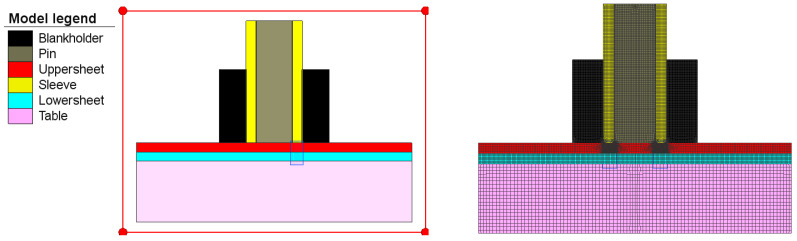
Finite element mesh of the 2D axisymmetric model.

**Figure 7 materials-16-04519-f007:**
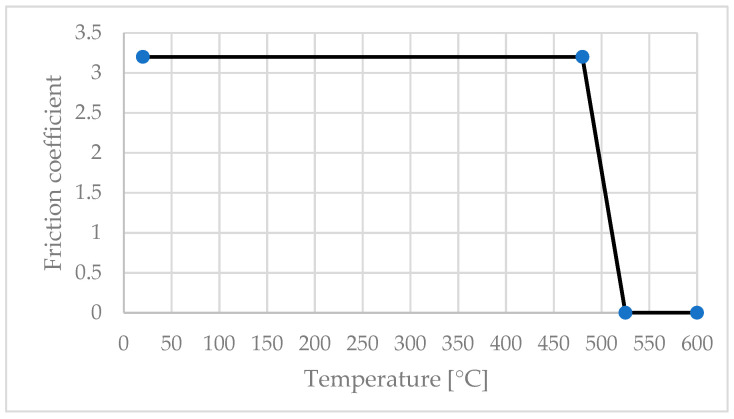
Variation of friction factor vs. temperature.

**Figure 8 materials-16-04519-f008:**
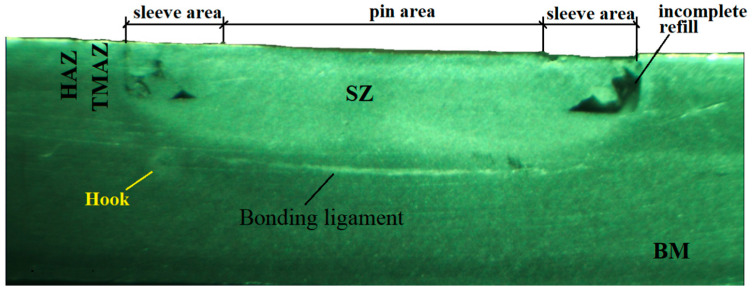
Cross-section through the welded joint.

**Figure 9 materials-16-04519-f009:**
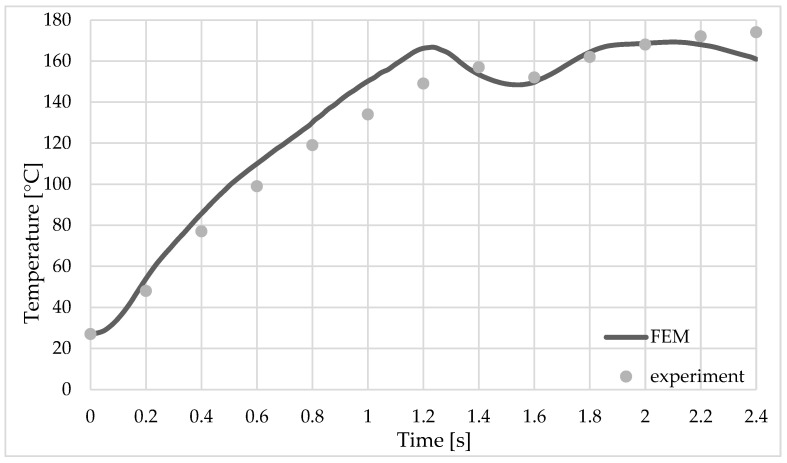
Temperature variation during the welding process at selected measuring point sensor_1.

**Figure 10 materials-16-04519-f010:**
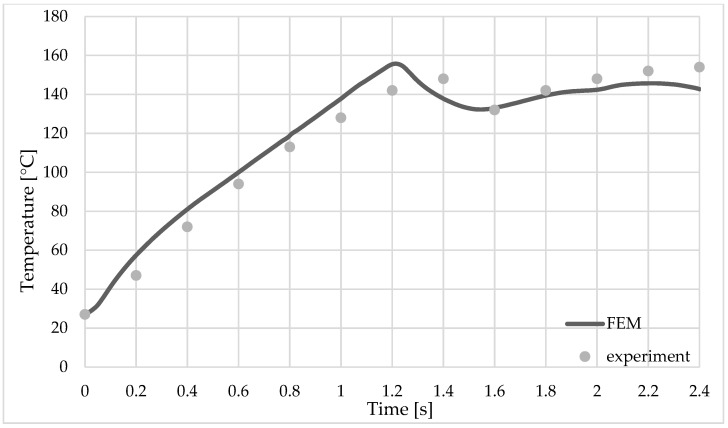
Temperature variation during the welding process at selected measuring point sensor_2.

**Figure 11 materials-16-04519-f011:**
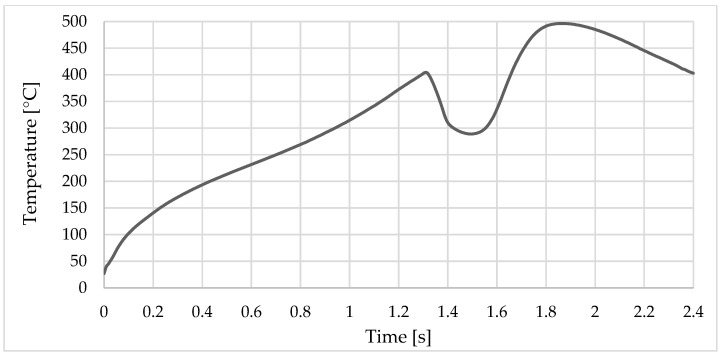
Temperature variation during finite element analysis.

**Figure 12 materials-16-04519-f012:**
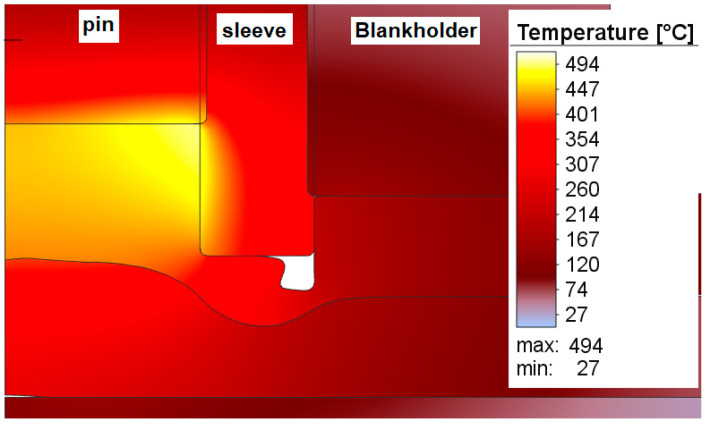
Temperature at the interface of the tool and the workpiece at the welding time corresponding to 1.8 s.

**Figure 13 materials-16-04519-f013:**
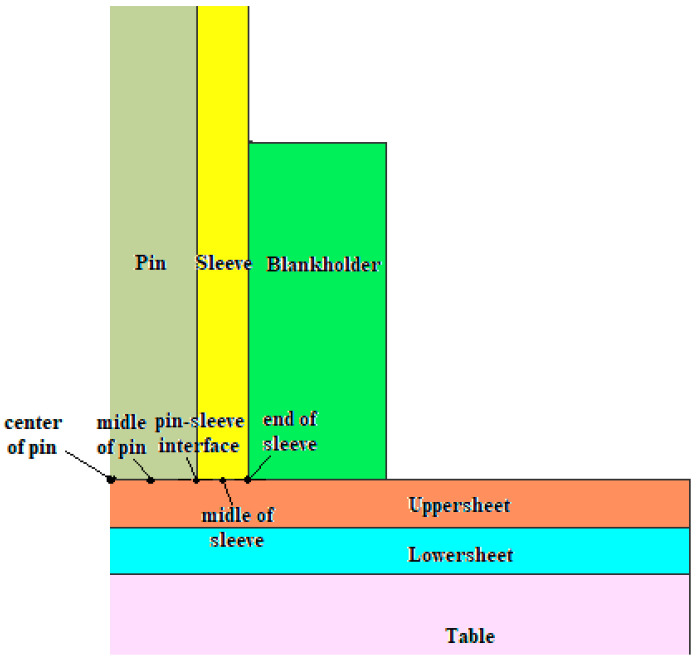
The points selected to illustrate the temperature variation over time.

**Figure 14 materials-16-04519-f014:**
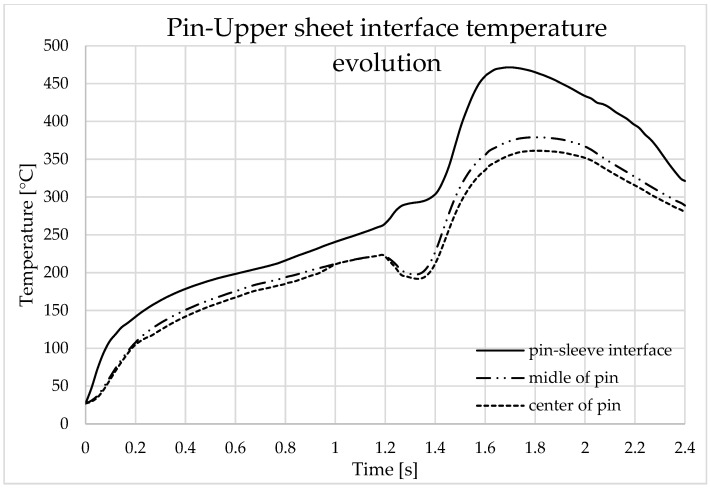
Temperature variation during finite element analysis at the pin–upper sheet interface.

**Figure 15 materials-16-04519-f015:**
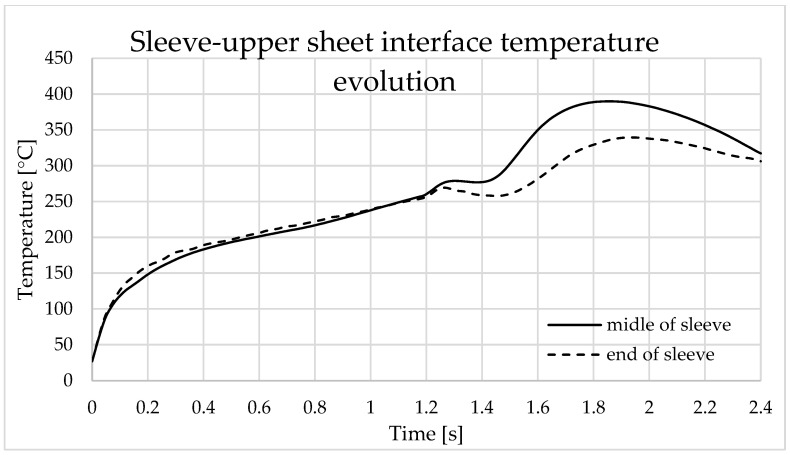
Temperature variation during finite element analysis at the sleeve–upper sheet interface.

**Figure 16 materials-16-04519-f016:**
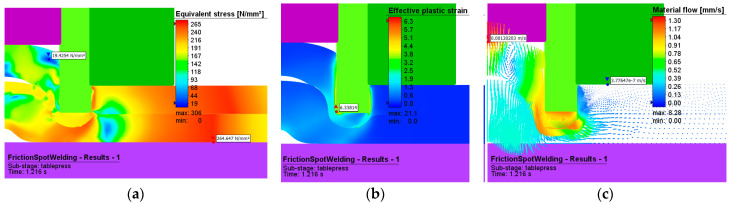
(**a**) Effective stress (MPa), (**b**) plastic deformation, and (**c**) material flow during the plunging stage.

**Figure 17 materials-16-04519-f017:**
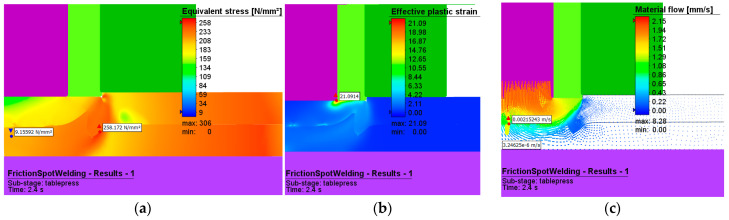
(**a**) Effective stress (MPa), (**b**) plastic deformation, and (**c**) material flow during the refilling stage.

**Figure 18 materials-16-04519-f018:**
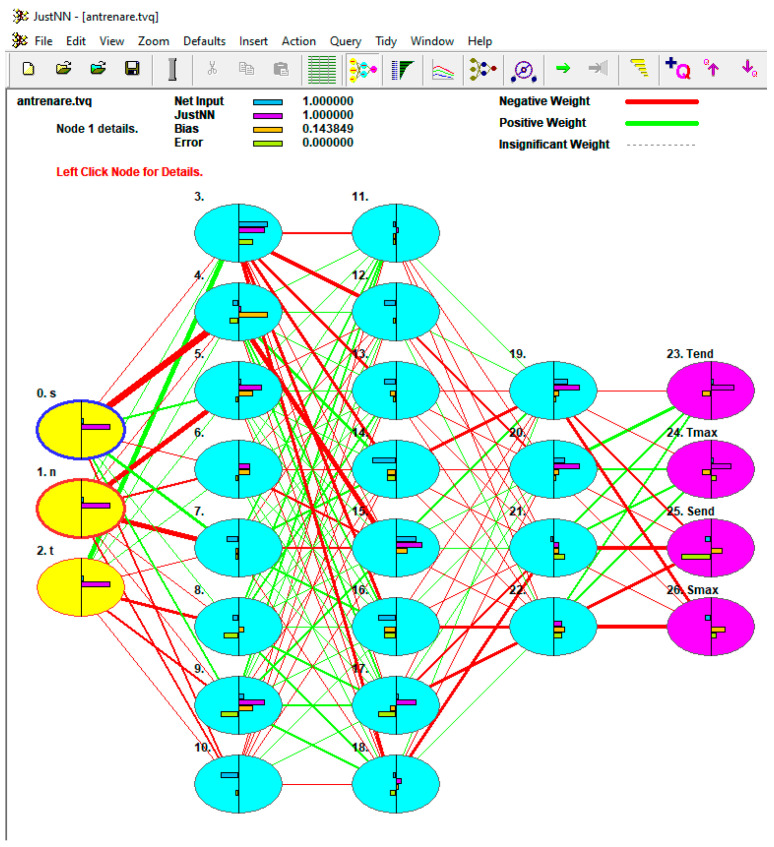
Graphical representation of the neural network.

**Figure 19 materials-16-04519-f019:**
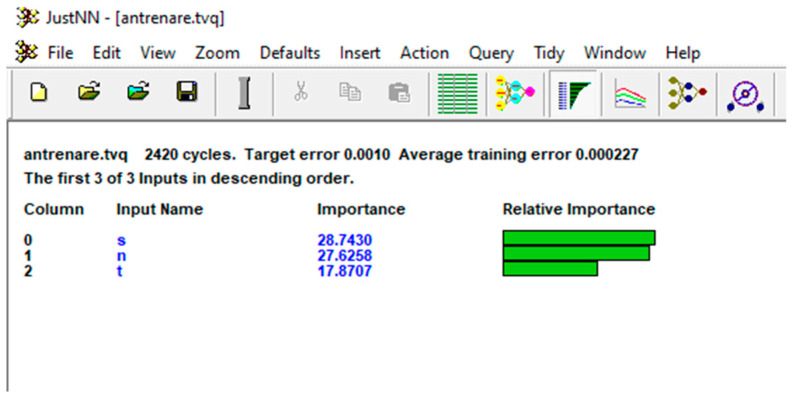
The input importance view.

**Figure 20 materials-16-04519-f020:**
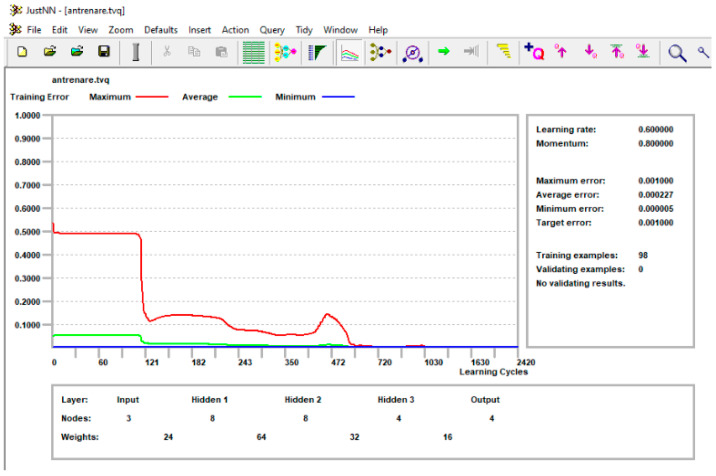
Neural network training.

**Table 1 materials-16-04519-t001:** Chemical composition for the AA6061 alloy (min–max %).

Al	Mg	Si	Fe	Cu	Cr	Zn	Ti	Mn
95.85	0.8	0.4	0.0	0.15	0.04	0.0	0.0	0.0
98.56	1.2	0.8	0.7	0.40	0.35	0.25	0.15	0.15

**Table 2 materials-16-04519-t002:** Basic mechanical properties of 6061-T6 aluminum alloy.

Yield Stress Rp_0.2_ (MPa)	Ultimate Tensile Stress *Rm* (MPa)	Elongation at Fracture *A*, %	Melting Point °C
276	310	12	582–652

**Table 3 materials-16-04519-t003:** Parameters in the plasticity model of base metal.

Mat.	C_1_	C_2_	l_1_	l_2_	m_1_	m_2_	n_1_	n_2_
AA 6061-T6	405.9	−0.0032	−4.344 × 10^−5^	0.01674	0.000373	−0.05181	−0.00056	0.253023

**Table 4 materials-16-04519-t004:** The result of comparing the output values provided by the neural network with the output values obtained by simulation.

T_end_	T_max_	σ_end_	σ_max_	T_end_ calc	T_max_ calc	σ_end_ calc	σ_max_ calc	Δ T_end_ [%]	ΔT_max_ [%]	Δσ_end_ [%]	Δσ_max_ [%]
406	497	270	317	406.137	495.232	268.830	317.810	−0.03	0.35	0.43	−0.25
381	480	270	313	383.366	480.131	268.219	312.998	−0.62	−0.03	0.66	0.00
432	520	251	301	431.486	518.651	249.919	302.910	0.12	0.26	0.43	−0.63
398	489	288	335	399.890	488.202	289.279	335.350	−0.47	0.16	−0.44	−0.10
373	472	274	317	372.572	471.614	273.617	316.964	0.11	0.08	0.14	0.01
417	505	260	310	417.046	504.487	257.778	309.612	−0.01	0.10	0.85	0.12
408	497	261	308	407.550	497.916	260.754	307.950	0.11	−0.18	0.09	0.02
369	468	275	318	370.521	469.810	274.924	317.469	−0.41	−0.39	0.03	0.17
413	501	261	311	411.86	499.954	259.416	311.096	0.27	0.21	0.61	−0.03
405	496	262	309	404.61	495.442	262.249	309.283	0.1	0.11	−0.09	−0.09

## Data Availability

Not applicable.

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
