# Peer review of "Neural Networks Applied for Predictive Parameters Analysis of the Refill Friction Stir Spot Welding Process of 6061-T6 Aluminum Alloy Plates"

_materials, 2023, doi:10.3390/ma16134519_

Round 1

Reviewer 1 Report

1. In Figure 5, the connection scheme of the thermocouple seems ambiguous. Please improve it.

2. In the cross-section the four zones, characterized by different microstructures, can be seen: the mixing zone (SZ), the thermo-mechanically affected zone (TMAZ), the heat affected zone (HAZ) and the base material (BM).However, in Figure 9, the author did not show these four regions. In addition, the quality of the cross-sectional macro image is poor and the displayed area is incomplete. Please replace it.

3. The article has Figures 19 and 21 but not Figure 20. Please check and add.

The article language has been slightly improved.

Author Response

The authors would like to express their gratitude to the reviewer for their competent criticism and constructive comments, which have improved the manuscript. We have given careful attention to each point raised by the reviewer and have made revisions to the manuscript based on their comments. The responses are as follows: the reviewer's comments are written in italics, the responses are written in regular font, and the manuscript changes are highlighted in red.
We thank the reviewer for the thorough input on our manuscript and we have addressed all the reviewer’s suggestions as described below.

Reviewer 2 Report

This is a timely effort by authors on "Neural networks applied for predictive parameters analysis of the refill friction stir spot welding process of 6061-T6 aluminum alloy plates". However, there are few suggestions to improve this manuscript. 

1. Novelty needs to be highlighted in a better way.

2. Fig. 1 must be referred if this does not belong to your creative work.

3. Fig. 2 is badly referred. A figure must be mentioned in the text first before one can see it. Here the reverse is true. This should be watched carefully throughout this manuscript.

4. No. of reference must be increased up to 35 within the time span of last five years. Currently, they are merely 24. 

5. No. of references were improperly cited. E.g. I could not find the reference no. 2 in the introduction section. Numbering gaps of references can clearly be observed.  After reference no. 6, 9 suddenly appears. 

6. I can not see the Figs. 3&4 mentioned in the text before theses figures appear in the text. 

7. Please explain thoroughly the Fig. 4. Currently, nothing I could find towards the Fig. 4.

8.  Figure 5 is badly captured. One can never see the k-type thermocouples in this figure.

9. Please explain sufficiently; how did you find empirically the parameters in Table 3.

10. Please mention the melting point of 6061-T6 Aluminum in Table 2.

11. There is 3rd line pattern i.e. dashed equally spaced line in Fig. 14&15. What does that line indicates?

12. Figure 17 is not mentioned in the text. What is the difference between Fig. 16 and 17?

13. Table 4 is full of jargons. Parameters of this table must be explained first especially the subscripts with them.

14. Future work must be provided at the end of the conclusion section. 

Extensive English Language editing is required as per the modern standard of English.

Author Response

(The authors gave the same response as above.)

Reviewer 3 Report

Authors used three input parameters and require significant attention on how these parameters are selected for experimentation, prediction and optimization.

Authors used FEM to get the data required for training and optimization. However, the major drawback in this research work is assumed data during FEM model development. The assumptions made during FEM simulations are difficult to ensure in the practical experiments.

Which algorithm is employed to train the neural network need to be explained in detail. It is better to compare 2-3 different algorithms and comparision would be more interesting.

The hidden layer and units’ selection is not clearly explained.

The network stopping criteria i.e. error goal, is not explained in detail.

The bias value, learning rate, and momentum constant values and  their selection criteria has not been explained in detail.

The pin material, and their dimensions, also influences greatly on the welding quality. However, authors have not studied their effects. Kindly explain.

The novelty of the present work need to be explained in detail.

SEM analysis if performed, will justify the supporting results. 

Note that there are few abbreviations need to be expand when it appears first in the revised manuscript. 

Author Response

(The authors gave the same response as above.)

Reviewer 4 Report

This manuscript presented decent amount of work in the parameters measurement and analysis of the refill friction stir spot welding process (RFSSW). The experimental design was effective. However, the writing included too much less relevant information. And the most controversial part of this manuscript is the purpose of the neural network training. The authors have mentioned in the introduction section that longer holding time and slower rotation speed will improve the weld properties but decrease the efficiency. It will be helpful to study how to resolve these conflicting impact. However, the whole experiments and neural network training didn't really provide useful guiding information for the modification of RFSSW. 

The English writing of this article is acceptable but can be improved with professional writing aid.

Author Response

(The authors gave the same response as above.)

Round 2

Reviewer 2 Report

Now the authors have the manuscript a lot.

The paper could be accepted in current form, however, the organization of manuscript should be altered in the form of Abstract (correct so far), Introduction (correct so far), Materials and Methords, Results and Discussion, and Conclusions (correct so far).

Author Response

The authors would like to express their gratitude to the reviewer for their competent criticism and constructive comments, which have improved the manuscript. We have given careful attention to each point raised by the reviewer and have made revisions to the manuscript based on their comments. The responses are as follows: the reviewer's comment is written in italics and the response is written in regular font.
